# Performance Differences of Temporal Parameters and Point Outcome Between Elite Men’s and Women’s Badminton Players According to Match-Related Contexts

**DOI:** 10.3390/ijerph16214057

**Published:** 2019-10-23

**Authors:** Miguel-Ángel Gomez, Fernando Rivas, Jonathan D. Connor, Anthony S. Leicht

**Affiliations:** 1Department of Social Sciences, Physical Activity, Sport and Leisure, Polytechnic University of Madrid, 28031 Madrid, Spain; fernandorivas77@gmail.com; 2Spanish Badminton Federation, 28040 Madrid, Spain; 3Sport and Exercise Science, James Cook University, Townsville 4814, Australia; jonathan.connor@jcu.edu.au (J.D.C.); anthony.leicht@jcu.edu.au (A.S.L.)

**Keywords:** racket sports, rally time, rest time, strokes

## Abstract

Background: the aim of the present study was to identify the temporal and notational performance differences between elite men’s and women’s badminton players according to match type and set. Methods: the sample was composed of 60 men’s and 60 women’s matches classified by match type or duration: short (lower quartile), long (upper quartile) and regular matches (interquartile range). Temporal and notational variables were analysed for each match and compared between sexes accounting for match duration. Results: greater intensity for most variables (i.e., rally time, rest time, density, and strokes per match/rally) was exhibited in men’s matches compared with women’s matches (i.e., higher frequency between strokes). In addition, the greater intensity for men compared to women was more pertinent during long matches (13 significant variables) and less evident during short matches (six significant variables). Point outcome displayed similar trends for each sex during matches with more winners for men’s players when serving and more unforced errors for women’s players. Lastly, based on each match type, fewer sex differences were noted during sets 2 and 3, as the match progressed. Conclusion: men’s players performed at a greater intensity than women’s players for different match and set contexts, with this sex difference enhanced when controlling for match type and set. The development of sex- and match-specific scenarios will assist coaches and trainers in the design of specific training drills to enhance the athletic performance of elite badminton players.

## 1. Introduction

Badminton is an intermittent and dynamic racket sport (i.e., combining speed, high intensity actions, anticipation and rest intervals), where both competing players have to act and react during the rallies (i.e., interaction of technique, tactics and decision making) to gain temporal and positional advantages over their opponents in order to win the point [1,2,3,4,5]. Nowadays, the current badminton regulations (i.e., scoring system) have resulted in changes for players’ performances with longer matches incorporating more aggressive playing styles, different tactical patterns and reduced time between strokes during longer rallies [3,6]. Abian-Vicen et al. [1] have suggested that badminton is evolving with an increase in the intensity of rallies, sets and matches. The authors argued that the scoring system has altered the players’ performance profiles with constant adaptations to a high-intensity and demanding game (i.e., more rallies played, more strokes per rally, more variability of rally time and rest time during rallies). Similar to other racket sports [7,8], modelling the performance of elite badminton players is needed to improve players’ profiles and represent the real competition contexts (stable and unstable scenarios). 

As a result of these evolving performance trends, most studies of badminton [9,10,11] have focused on identifying temporal and notational characteristics of men’s and women’s players during elite competitions. Abian-Vicen et al. [12] analysed 20 singles matches (only those played with two sets) from the Beijing Olympic Games (2008) (*n* = 10 matches for each sex), with total match duration, rally time, rest time, number or rallies, strokes per rally and shot frequency greater for men’s compared to women’s players. On the contrary, women’s players performed at a higher work density and played longer match times than men’s players. The authors did not find temporal variable differences between set 1 and 2. In addition, shorter rally times ranged between 3 and 6 s and were more frequent for women’s players, while longer rally times (>16 s) were more frequent for men’s players [12]. Subsequently, Gawin, et al. [10] investigated the performance characteristics of each badminton discipline (singles and doubles for each sex, and mixed doubles) during 50 matches played by the World top 10 players. These authors confirmed the previous findings of Abian-Vicen et al. [1,12] where men’s players performed at a higher intensity, while women’s players performed at a greater workload and longer playing time, compared to the opposite sex. More recently, Valldecabres, de-Benito, Casal, and Pablos [13] studied the differences between men’s and women’s singles matches during the 2015 World Badminton Championship. Their main findings indicated that men’s players experienced greater total match duration, real time played, effective time of play (%), rally time, strokes per rally, rest time, rest time at point interval (rest time at 11th point), and strokes per match than women’s players [13]. However, women exhibited a greater work density, frequency of strokes, and rest time between sets than men’s players [13]. More recently, Torres-Luque, Fernandez-Garcia, Blanca-Torres, Kondric, and Cabello-Manrique [14] studied the differences between sets in all the disciplines of badminton (singles, doubles and mixed doubles) for both sexes. Their results were in accordance with the previous studies cited above. However, a great disparity exists for the match duration of elite badminton studies with these ranging from 40 to 50 min for men and 17 to 28 min for women [1,8,13,14]. Establishing a distinctive approach that accounts for match duration and sets when examining the temporal parameters of men’s and women’s badminton matches would provide more realistic performance profiles for men’s and women’s players during long (three sets) and short (two sets) matches [14]. Previous studies in badminton [10,11,12,13,14] have not accounted for match type and set characteristics when examining sex differences in performances. Prior studies in other racket sports such as tennis [15,16,17], table tennis [18,19,20], squash [21,22] or paddle [23,24] have considered these crucial factors and emphasised the importance of analysing players´ performances according to match-related contexts. Therefore, the aim of the present study was to investigate the effect of sex on temporal parameters and notational variables (e.g., range of strokes and point outcome) in elite badminton accounting for match type and set. Our hypothesis was that men’s players would perform with greater intensity than women’s players for different match and set contexts. Clarification of sex differences during specific match scenarios (i.e., match type and sets) would assist coaches in the development of sex- and scenario-specific training and match strategies for success in elite badminton [13,14].

## 2. Materials and Methods 

### 2.1. Sample

The sample was obtained from 60 men’s and 60 women’s matches (video recorded matches from the World Badminton Federation) played during the 2016 World Badminton Super Series. In order to control for the variability of match performances based on time in elite badminton [11], total match time was identified with the median, and upper and lower quartiles employed to establish three distinct types of matches: regular matches (within the interquartile range, *n* = 20 for each sex), short matches (below lower quartile, *n* = 10 matches for each sex) and long matches (above upper quartile, *n* = 10 matches for each sex). All data were obtained from a public site in accordance with the European Data Protection Law, and the current study was conducted with the approval of the Polytechnic University of Madrid Ethics Committee.

### 2.2. Procedure

The matches were examined by four experienced badminton coaches (10 years of experience as elite badminton coaches) using the video analysis program Dartfish (Friburgo, Switzerland), with match temporal parameters and point outcome being the key variables. Reliability (inter- and intra-observer) for match observations were assessed as good and very good (Kappa ≥ 0.81; Pearson’s correlation coefficient: r > 0.86; intraclass correlation coefficient (ICC) ≥ 0.85, and the typical error of measurement (TEM) ≤ 0.46) using 12 randomly selected matches [25,26].

The independent variable was sex, while the dependent variables for each match were considered according to the scientific available research [1,2,3,4,5,6,7,8,9,10,11,12,13,14,27,28]: match/set duration, rally time, real playing time, rest time, real rest time, density, number of strokes per match/set, number of strokes per rally, frequency of strokes, number of rallies, effective playing time (%) and effective rest time (%), number of strokes per rally and rally outcome. In addition, point outcome (winning the rally as the player serving or receiving; and the type of action performed = winning stroke, forced error or unforced error) and range of strokes per rally (considering 2 strokes per range where each player performs one stroke each: A: 1–2; B: 3–4; C: 5–6; D: 7–8; E: 9–10; F: 11–12; G: 13–14; H: 15–16; I: 17–18; J: 19–20; and K: >20) were also considered as notational analysis variables. 

### 2.3. Statistical Analysis

Firstly, the Shapiro–Wilks test was used to check data normality assumptions for all temporal parameters (*p* < 0.05) followed by the use of the Student’s t-test to analyse sex differences for variables and each match context. Cohen’s d effect size (ES) was calculated and interpreted via the following ranges: 0.20 = small effect, 0.50 = medium effect, and 0.80 = large effect [29]. Secondly, relationships between sex, and range of strokes and point outcome notational variables were determined using the Pearson’s Chi-square test (Crosstabs Command) with adjusted residuals (ARs) > 1.96 classified as significant. Additionally, the Cramer’s V test was conducted to estimate ES with interpretation based on the following values: 0.10 = small effect, 0.30 = medium effect, and 0.50 = large effect [30]. Statistical analyses were undertaken using the statistical software IBM SPSS for Windows version 25 (IBM. Corp. Armonk, NY, USA) and significance level was set to 0.05.

## 3. Results

The descriptive and univariate differences between men’s and women’s players for all temporal parameters are shown in Table 1. Significant sex differences (*p* < 0.05) were identified with women’s players exhibiting a greater frequency of strokes (medium ES) and less rally time, rest time, density, strokes per rally, and strokes per match (small–medium ES) compared to men’s players. Significant relationships were identified between the range of strokes and sex (Table 1, small ES) with higher frequencies of short ranges (A and B) for women’s players and higher frequencies of long ranges (H, I and K) for men’s players.

According to each match type, significant differences (*p* < 0.05) between sexes were noted for short matches, with men’s players exhibiting greater rest time and effective rest time (%) (large ES) and lower density and frequency of strokes (large ES) compared to women’s players (Table 2). Significant relationships were identified between the range of strokes and sex (small ES = 0.15), with men’s players performing more long rallies (range K) compared to women’s players (Table 2). For regular matches, men’s players experienced a greater number of strokes per match (large ES), and greater match duration, rally time, real playing time, density and strokes per rally (medium ES) compared to women’s players (Table 2). Significant relationships were identified between the range of strokes (small ES) and sex with more short (range A) and long rallies (range K) performed by women’s and men’s players, respectively (Table 2). For long matches, men’s players experienced significantly greater rally time (large ES), density (large ES), strokes per rally (small ES), strokes per match (large ES) and effective playing time (large ES), compared to women’s players (Table 2). In contrast, women’s players experienced a significantly greater frequency of strokes (large ES) and number of rallies (small ES, Table 2). Significant relationships were identified between sexes (small ES) with more short (range A) and long rallies (range K) performed by women’s and men’s players, respectively (Table 2).

Results for short (Table 3), regular (Table 4) and long (Table 5) matches with consideration of set have been provided. During short matches, significant differences between sexes were noted for set 1 with men’s players experiencing greater real rest time, strokes per set, number of rallies and effective playing time (all large ES) compared to women’s players (Table 3). In contrast, women’s players experienced greater real playing time, frequency of strokes and effective rest time (all large ES) compared to men’s players (Table 3). During set 2, men’s players experienced greater real rest time) and strokes per set (all large ES) compared to women’s players (Table 3). No significant relationships between the range of strokes and sex were noted during either set (Table 3).

During regular matches, significant differences between sexes were noted with men’s players experiencing greater real playing time (large ES) and strokes per match (large ES) during all sets (Table 4). Men’s players also experienced greater real rest time (large ES) during sets 2 and 3 and greater set duration (large ES) during set 3 (Table 4). Significant relationships between the range of strokes and sex were identified for sets 1 and 2 (all small ES) with more short (range A) and long (range K) rallies performed by women’s and men’s players, respectively (Table 4).

During long matches, significant differences between sexes were noted with men’s players experiencing greater real playing time, real rest time and strokes per match (all large ES) during sets 1 and 2 compared to women’s players (Table 5). Men’s players also experienced greater effective play time (large ES) during set 1 compared to women’s players who experienced greater effective and real rest time (large ES) during sets 1 and 3, respectively (Table 5). Significant relationships between the range of strokes and sex were identified during sets 1 and 2 (small ES) with more long rallies (range K) performed by men’s players compared to women’s players, while women’s players performed more short rallies (range A) during set 1 and 3 compared to men’s players (Table 5).

Results of sex differences for point outcome are presented in Table 6. Significant relationships were identified for: (i) all matches (small ES), with more points won via winners (19.1% vs. 15.1%) for men’s players when serving, with more points won via forced errors (15.5% vs. 13.8%); for women’s players when serving, with more points won via unforced errors (21.6% vs. 18.1%) for women’s players when receiving; (ii) short matches, with more points won via unforced errors (19.7% vs. 14.4%) for women’s players when serving; (iii) regular matches (small ES), with more points won via winners (19.0% vs. 15.0%) for men’s players when serving and more points won via unforced errors (22.2% vs. 17.4%) for women’s players when receiving; and (iii) long matches (small ES), with more points won via winners (18.9% vs. 13.7%) for men’s players when serving and with more points won via forced errors (16.6% vs. 13.3%) for women’s players when serving

Results of set differences for point outcome are presented in Table 7. For regular matches, significant relationships were identified during: (i) set 1 (small ES), with more points won via winners (19.6% vs. 13.5%) for men’s players when serving while more points were won via forced errors (17.2% vs. 12.6%) and unforced errors (22.4% vs. 18.0%) for women’s players when serving and receiving, respectively; and (ii) sets 2 and 3 (small ES), with more points won via unforced errors (21.3% vs. 17.3%; and 26.1% vs. 15.3%, respectively) for women’s players when receiving (Table 7). Lastly, for long matches, significant relationships between sexes were noted during set 1 (small ES), with more points won via winners (22.9% vs. 16.2%) for women’s players when receiving (Table 7).

## 4. Discussion

The aim of the study was to investigate the sex differences on temporal parameters and notational variables (range of strokes and point outcome) depending on match type and set in elite badminton matches. Similar to prior studies, men’s players performed at higher intensities than women’s players for most of the temporal parameters analysed during all matches (e.g., rally time, rest time or number of strokes) [1,3,9,10]. However, the analysis based on match type and set allowed the identification of different trends and performance profiles related to each sex in those contexts (e.g., short, regular or long matches). In addition, the notational variables (i.e., point outcome and range of strokes) showed different performance profiles for men’s and women’s players in each match and set, highlighting the importance of considering performance variability related to context factors [14]. The current study has added to the knowledge base for racket sports, with coaches encouraged to consider sex differences and their associated controlling match-related contexts [7,8] during the development of training tasks.

Irrespective of match length, a number of temporal parameters were greater for males (i.e., rally time, rest time, density, strokes per rally, strokes per match/set), albeit lower frequency between strokes compared to females (see Table 8). These findings could be related to men´s physical capacities and strength that allow them to perform with greater intensity and explosiveness during rallies compared to women´s players [12]. Males also had a consistently greater range of strokes (e.g., H: 15–16, I: 17–18, and K: >20), while females demonstrated increased short-range strokes (e.g., A: 1–2, B: 2–3). In fact, men´s players were able to maintain the physical intensity with technical effectiveness during the rallies due to their experiences with quick and high-demand rallies that required anticipation and explosive movements [14]. Finally, men had a greater amount of serving winners, while women had greater serving forced errors and receiving unforced errors. These results support those of Avian et al. [12] who reported that men´s players played with higher stroke frequency and explosive actions (e.g., Smash) surprising the opponents and finishing the point with a winner. These results indicate the importance of rest time and recovery between points that may impact on forced and unforced errors (with greater effect for women´s players during long and regular matches) [10,12,14]. Additionally, these findings were consistent with previous literature in badminton [1,3,9,10], whereby men’s matches had a tendency to be longer in duration due to an increased number of certain temporal parameters (e.g., rally time, rest time, density, strokes per rally, strokes per match/set). 

A number of significant differences were evident between both sexes, and the match type and sequence of sets. During shorter matches, there was a tendency for an increase in real playing time and real rest time across both sets for men, in comparison to women’s matches. There was also a greater propensity of strokes per match/set, number of rallies, and effective playing time (%), albeit only during the first set of men’s short matches. These trends were previously identified for badminton matches reflecting the increase of physical intensity during longer matches/sets of authentic competitions [1,3,12,14]. Additionally, there was an overall increase in the number of strokes per set, rallies and effective playing time (%), while women had a greater frequency of strokes and effective rest time. During regular length matches, men displayed greater real playing time, strokes per match/set, while women had greater, receiving unforced errors. During the early sequence of sets (e.g., sets 1 and 2), men demonstrated a greater range of strokes (K: >20) and served more winners, while during the latter sets, there was an increase in rally time and rest time. This result likely explains the reduced number of received unforced errors, as this reduction would likely lead to longer playing time. According to Chiminazzo et al. [31], higher temporal demands of long rallies required an increase of total rest time for athletes. Thus, elite badminton players may be more aware of spending more rest time between points and slowing down the playing style or rhythm of the last set (i.e., effective recovery and better preparation for the next point) [32]. Finally, during long matches, rally time and real playing time were significantly greater for men’s competition. Men also had a greater number of strokes per rally, strokes per match/set, and effective playing time (%), while women reported greater density, frequency of strokes, and effective rest time (%). Women also had a shorter range of strokes and serving of forced errors in comparison with men’s matches, who had a greater range of stokes and serving winners. These results support the suggestion that badminton player´s performances evolve from a predominant quantitative (e.g., longer matches and higher intensity during set 3) to a qualitative (e.g., an increase of unforced errors in set 3 and long matches) focus [14]. In particular, the temporal requirements were most relevant during long matches, but as the match progressed, the frequency of strokes increased with the execution effectiveness compromised by fatigue [14]. Together, these findings highlight the differing temporal parameters, range of strokes, and point outcomes between men’s and women’s competition with important consideration of the duration of matches and the sequence of sets. Therefore, consideration of these current trends within matches should be undertaken by coaches and trainers to improve training plans and competition management accounting for match type and set.

The current study enabled the novel exploration of temporal and notational demands of men’s and women’s, elite badminton players while considering performance variability of match-related contexts (match type and set). However, some limitations of the current study should be noted and considered for future studies. The temporal and notational variables examined in the current study were based on prior work [1,2,3,4,5,6,7,8,9,10,11,12,13,14]. However, others including technical- (e.g., type of serve or technical skill used to end the rally), tactical- (e.g., strokes zones distribution for both sexes), or contextual-related (i.e., competition stage and quality of opposition) variables could be examined for a greater understanding of elite athletic performance. The current study included matches from the same tournament, with an examination of performance evolution over multiple tournaments (e.g., Olympic Games since Beijing 2008 to Tokyo 2020) likely providing a comprehensive reflection of match performance trends for each sex. Indeed, further research might consider a multivariate approach to define all possible real match-scenarios for men’s and women’s competition to support staff (coaches, trainers, etc) in the development of athletes with specialised preparation.

## 5. Practical Applications

The current findings have significant practical implications for coaches and players for their preparation of various match scenarios [14,19]. A tailored approach using real match demands for men’s and women’s players would offer a better foundation for the development of integrated conditioning plans [31,32]. Specifically, men’s and women’s players perform differently during short, regular or long matches, with the current reference temporal parameters encouraged for use in the design of training drills (i.e., rally and rest time, density and strokes per rally), controlling for total playing time (real playing time and effective playing time) and each match type and set. In addition, the use of a key variable such as the range of strokes (a minimum number of strokes to play short rallies in women’s players and a greater number of strokes during long rallies in men’s players) may improve the requirements for each drill and minimise unforced errors under fatigue conditions. Subsequently, we would suggest that training focus equally on total time duration of regular matches (50% of training time), and the needs and requirements of short and long matches (50% of training time). For example, the specific short/long match training could include specific tasks and/or worst case scenarios such as quick rallies (one to two strokes for women; one to four for men), long rallies (longer than 18 strokes for men; longer than 20 strokes for women), sequences of short and high-intensity rallies that accumulate fatigue (rallies with eight strokes and 6 seconds of time duration for men’s players; 10 strokes and 8 seconds for women´s players), and simulation of critical match instants affected by fatigue such as the last rallies of a close match during set 3 (rally duration of 11–12 seconds and with density of lower than 0.40) or the last point of each set and interval where decision making is potentially compromised (e.g., players use winners to end the rally quickly).

## 6. Conclusions

In conclusion, the current study of badminton performance has confirmed that men’s players perform at a greater intensity than women’s players for different match and set contexts. Specifically, greater intensity for temporal parameters was identified in men’s matches (i.e., rally time, rest time, density, and strokes per match/rally) compared with women’s matches (i.e., higher frequency between strokes). In addition, the sex differences varied according to match type, with greater intensity in men’s matches more relevant during long matches (e.g., significant differences in nine temporal parameters), and less evident during short matches (e.g., significant differences in five temporal parameters). In contrast, the point outcome displayed similar trends for each sex during matches with more winners for men’s players when serving and more unforced errors for women’s players. Lastly, based on each match type, fewer sex differences were noted during sets 2 and 3, as the match progressed. Further, the sex differences for elite badminton players were enhanced when controlling for match type and set highlighting the importance of these factors during performance analyses in badminton. Clarification of sex differences during specific badminton match scenarios will assist coaches and trainers to design specific training drills that simulate the current temporal and notational profiles of real match types.

## Figures and Tables

**Table 1 ijerph-16-04057-t001:** Differences in temporal parameters and range of strokes between men’s and women’s badminton players.

Variables	All Matches				*p*-Value	ES
	Men		Women			
Temporal Parameters	M	SD	M	SD		
Match duration	49.14	18.04	46.95	19.57	0.68	0.12
Rally time	9.91	8.03	9.05	6.55	<0.01 *	0.13
Real playing time	13.60	5.18	12.19	4.51	0.12	0.29
Rest time	22.94	12.72	22.04	11.91	0.01 *	0.07
Real rest time	29.57	11.60	28.04	12.62	0.50	0.13
Density	0.45	0.29	0.43	0.26	0.01 *	0.07
Strokes per rally	10.63	8.28	8.73	6.58	<0.001 *	0.25
Strokes per match	880.7	318.3	714.6	266.7	0.01 *	0.57
Frequency of strokes	0.90	0.13	1.06	0.30	<0.001 *	0.69
Number of sets	2.32	0.47	2.26	0.44	0.54	0.13
Number of rallies	82.89	20.17	80.64	19.33	0.64	0.11
Effective playing time (%)	31.68	3.69	31.11	4.49	0.47	0.14
Effective rest time (%)	68.32	3.69	68.89	4.49	0.47	0.14
**Range of strokes**	%	n	%	n	χ^2^	AR
A (1–2)	5.7	253	10.9	544		9.2 †
B (3–4)	17.6	787	19.2	960		2.1 †
C (5–6)	15.7	703	16.6	826		1.1
D (7–8)	13.3	594	14.0	700		1.1
E (9–10)	9.8	440	10.1	506		0.5
F (11–12)	8.2	366	7.8	387	174.86	0.8
G (13–14)	5.8	259	5.5	274		0.6
H (15–16)	5.3	236	3.9	196		3.1 †
I (17–18)	4.4	195	3.2	157		3.1 †
J (19–20)	3.0	134	2.6	128		1.3
K (21+)	11.4	508	6.3	312		8.8 †
Total number of ranges	4475		4990			
*p* value	<0.001 *					
ES	0.14					

* *p* < 0.05; † adjusted residual (AR) ≥ |2.0|; ES = effects size.

**Table 2 ijerph-16-04057-t002:** Differences in temporal parameters and range of strokes between men’s and women’s badminton players according to match type.

Variables	Short Matches						Regular Matches						Long Matches					
	Men		Women		*p*	ES	Men		Women		*p*	ES	Men		Women		*p*	ES
Temporal Parameters	M	SD	M	SD			M	SD	M	SD			M	SD	M	SD		
Match duration	29.01	3.21	26.95	3.63	0.19	0.60	45.39	6.85	41.89	7.16	0.04 *	0.50	82.07	5.05	83.54	10.50	0.69	0.18
Rally time	8.06	2.44	8.15	0.73	0.71	0.05	9.59	1.52	8.67	1.30	0.01 *	0.65	12.06	1.23	10.54	1.86	0.07	0.96
Real playing time	508.2	130.0	524.9	56.7	0.91	0.17	753.3	140.3	665.46	135.4	0.01 *	0.64	1367.3	143.3	1206.5	228.2	0.04 *	0.84
Rest time	16.70	1.64	14.10	2.27	0.01 *	1.31	22.35	2.39	21.35	2.53	0.05	0.41	28.55	2.08	28.12	4.44	0.96	0.12
Real rest time	998.4	130.8	856.3	173.6	0.06	0.92	1655.1	230.6	1541.0	258.7	0.08	0.47	3062.9	192.3	3054.6	550.7	0.78	0.02
Density	0.48	0.12	0.60	0.12	0.04 *	1.00	0.44	0.06	0.41	0.06	0.04 *	0.50	0.42	0.05	0.37	0.03	0.01 *	1.21
Strokes per rally	8.27	6.89	6.71	4.39	0.06	0.27	10.33	7.74	8.59	6.47	0.01 *	0.25	12.62	9.55	10.13	7.52	0.01 *	0.29
Strokes per match	528.5	128.6	432.8	68.2	0.06	0.93	822.2	141.5	662.6	139.2	0.01 *	1.37	1431.7	136.8	1160.0	196.9	0.01 *	1.60
Frequency of strokes	0.92	0.04	1.08	0.09	0.01 *	2.30	0.90	0.03	0.99	0.07	0.49	1.67	0.93	0.03	1.01	0.06	0.01 *	1.69
Number of rallies	63.90	6.19	64.50	5.50	0.65	0.10	78.47	14.68	75.38	12.66	0.06	0.23	113.4	3.69	114.4	5.94	0.03 *	0.20
Effective playing time (%)	33.57	5.42	38.50	4.61	0.08	0.98	31.20	2.85	30.07	2.80	0.42	0.40	30.95	2.43	28.36	1.76	0.02 *	1.22
Effective rest time (%)	66.43	5.42	61.50	4.61	0.02 *	0.98	68.80	2.85	69.93	2.80	0.49	0.40	69.05	2.43	71.64	1.76	0.67	1.22
Range of strokes	%	n	%	n	χ^2^	AR	%	n	%	n	χ^2^	AR	%	n	%	n	χ^2^	AR
A (1–2)	9.4	60	11.9	77		1.0	5.1	138	11.3	349		5.5 †	4.9	55	9.4	118		2.8 †
B (3–4)	23.2	148	24.5	158		0.3	18.1	488	19.5	602		0.9	13.3	151	15.9	200		1.1
C (5–6)	18.9	121	22.3	144		0.9	16.0	432	16.8	520		0.5	13.2	150	12.9	162		0.2
D (7–8)	13.9	89	16.4	106		0.8	13.8	373	13.4	415		0.3	11.6	132	14.2	179		1.2
E (9–10)	9.9	63	8.7	56		0.5	10.0	269	10.2	316		0.2	9.5	108	10.7	134		0.6
F (11–12)	6.6	42	5.7	37	27.7	0.4	8.4	226	7.7	238	117.5	0.6	8.6	98	8.9	112	56.0	0.1
G (13–14)	4.1	26	4.0	26		0.0	5.9	160	5.4	166		0.6	6.4	73	6.5	82		0.1
H (15–16)	4.1	26	2.2	14		1.4	5.1	138	3.9	121		1.5	6.3	72	4.8	61		1.1
I (17–18)	3.8	24	1.4	9		1.9	4.3	116	3.3	103		1.3	4.9	55	3.6	45		1.0
J (19–20)	2.3	15	1.7	11		0.6	3.0	81	2.5	76		0.8	3.4	38	3.3	41		0.1
K (21+)	3.9	25	1.1	7		2.3 †	10.4	281	5.9	181		4.2 †	17.8	202	9.9	124		3.6 †
Total number of ranges	639		645				2.702		3.087				1.134		1.288			
*p*	<0.001 *						<0.001 *						<0.001 *					
ES	0.15						0.14						0.15					

* *p* < 0.05; † AR = |2.0|.

**Table 3 ijerph-16-04057-t003:** Differences in temporal parameters and range of strokes between men’s and women’s badminton players during short matches.

Variables	Set 1						Set 2					
	Men		Women		*p*	ES	Men		Women		*p*	ES
Temporal Parameters	M	SD	M	SD			M	SD	M	SD		
Set duration	13.24	1.79	12.55	1.15	0.31	0.46	14.87	2.66	12.63	4.44	0.19	0.61
Rally time	7.85	6.09	8.39	4.63	0.83	0.10	8.05	7.63	7.91	4.45	0.96	0.02
Real playing time	244.87	5.68	258.54	4.50	0.01 *	2.67	263.36	6.11	266.42	4.24	0.21	0.58
Rest time	15.74	6.30	14.05	4.80	0.51	0.30	17.55	9.70	14.25	8.74	0.43	0.36
Real rest time	459.60	6.01	404.58	4.46	<0.01 *	10.4	538.82	9.24	451.74	6.09	<0.01 *	11.1
Density	0.51	0.32	0.63	0.32	0.41	0.38	0.49	0.32	0.61	0.31	0.41	0.38
Strokes per rally	8.13	6.26	6.81	4.43	0.59	0.24	8.41	7.51	6.61	4.36	0.52	0.29
Strokes per set	253.60	55.81	209.90	30.73	0.04 *	0.97	274.90	89.49	209.90	30.73	0.04 *	0.97
Frequency of strokes	0.91	0.14	1.18	0.35	0.03 *	1.01	0.90	0.14	1.14	0.34	0.06	0.92
Number of rallies	34.79	4.92	30.80	2.25	0.03 *	1.04	32.70	4.76	33.70	3.40	0.59	0.24
Effective playing time (%)	65.21	4.92	39.10	3.55	<0.01 *	6.08	32.34	6.64	37.91	6.18	0.07	0.87
Effective rest time (%)	34.79	6.09	60.90	3.55	<0.01 *	6.08	67.66	6.64	62.09	6.18	0.07	0.87
Range of strokes	%	n	%	n	χ^2^	AR	%	n	%	n	χ^2^	AR
A (1–2)	9.9	31	12.3	38		0.6	8.9	29	11.6	39		0.8
B (3–4)	21.8	68	23.4	72		0.3	24.5	80	25.5	86		0.2
C (5–6)	18.6	58	22.4	69		0.7	19.3	63	22.3	75		0.6
D (7–8)	14.4	45	16.6	51		0.5	13.5	44	16.3	55		0.7
E (9–10)	11.2	35	8.1	25		0.9	8.6	28	9.2	31		0.2
F (11–12)	7.1	22	4.5	14	14.17	0.9	6.1	20	6.8	23	23.66	0.3
G (13–14)	3.8	12	4.9	15		0.4	4.3	14	3.3	11		0.5
H (15–16)	3.8	12	3.2	10		0.3	4.3	14	1.2	4		1.7
I (17–18)	2.9	9	2.3	7		0.3	4.6	15	0.6	2		2.3
J (19–20)	2.6	8	1.6	5		0.6	2.1	7	1.8	6		0.2
K (21+)	3.8	12	0.6	2		0.9	4.0	13	1.5	5		1.4
Total number of ranges	312		308				327		337			
*p*	0.17						0.01					
ES	0.15						0.19					

* *p* < 0.05.

**Table 4 ijerph-16-04057-t004:** Differences in temporal parameters and range of strokes between men’s and women’s badminton players during regular matches.

Variables	Set 1						Set 2						Set 3					
	Men		Women		*p*	ES	Men		Women		*p*	ES	Men		Women		*p*	ES
Temporal Parameters	M	SD	M	SD			M	SD	M	SD			M	SD	M	SD		
Set duration	18.84	3.09	18.69	2.08	0.85	0.06	18.91	3.21	18.73	3.55	0.86	0.05	20.27	2.44	17.18	1.76	<0.01 *	1.45
Rally time	9.56	7.12	8.69	6.22	0.68	0.13	9.39	7.71	8.60	6.36	0.73	0.11	9.48	7.45	8.37	6.51	0.62	0.16
Real playing time	338.4	6.80	314.1	6.12	<0.01 *	3.75	332.9	7.43	301.2	5.91	<0.01 *	4.72	348.4	7.35	288.0	6.04	<0.01 *	8.98
Rest time	20.91	9.74	20.37	7.44	0.84	0.06	23.13	13.02	22.15	11.68	0.80	0.08	22.39	14.35	20.13	8.41	0.55	0.19
Real rest time	697.9	8.98	695.1	6.88	0.28	0.35	774.2	11.60	731.6	9.59	<0.01 *	4.00	777.9	13.94	652.8	8.01	<0.01 *	11.0
Density	0.47	0.29	0.43	0.26	0.65	0.15	0.43	0.29	0.40	0.23	0.72	0.12	0.42	0.23	0.43	0.31	0.91	0.04
Strokes per rally	10.39	7.56	8.58	6.36	0.42	0.26	10.30	8.06	8.57	6.37	0.46	0.24	10.30	7.61	8.63	6.68	0.47	0.23
Strokes per match	367.8	73.9	310.1	49.3	0.01 *	0.92	365.4	63.12	300.3	69.04	0.01 *	0.98	378.5	52.3	297.0	85.78	<0.01 *	1.14
Frequency of strokes	0.90	0.12	1.05	0.31	0.05	0.64	0.89	0.13	1.03	0.29	0.06	0.62	0.89	0.12	0.96	0.25	0.27	0.36
Number of rallies	35.38	3.22	36.13	2.73	0.43	0.26	35.47	3.91	35.03	3.87	0.72	0.11	36.75	4.46	34.43	3.95	0.09	0.55
Effective playing time (%)	32.71	3.17	31.11	2.56	0.35	0.56	30.36	4.05	29.18	3.45	0.33	0.31	30.95	1.75	30.22	4.88	0.53	0.20
Effective rest time (%)	67.29	3.17	68.89	2.56	0.35	0.56	69.64	4.05	70.82	3.45	0.33	0.31	69.05	1.75	69.78	4.88	0.53	0.20
Range of strokes	%	n	%	n	χ^2^	AR	%	n	%	n	χ^2^	AR	%	n	%	n	χ^2^	AR
A (1–2)	4.3	52	12.2	176		4.6 †	5.8	70	9.9	139		2.5 †	5.4	16	14.1	34		2.4
B (3–4)	19.0	228	18.8	271		0.1	17.8	215	20.7	290		1.1	15.3	45	17.0	41		0.4
C (5–6)	15.3	184	16.9	244		0.7	16.5	199	16.8	235		0.1	16.7	49	17.0	41		0.1
D (7–8)	13.0	156	13.6	196		0.3	14.7	177	13.3	186		0.7	13.6	40	13.7	33		0.0
E (9–10)	9.6	115	9.8	142		0.1	10.0	120	10.9	153		0.5	11.6	34	8.7	21		0.8
F (11–12)	9.3	112	7.7	111	72.05	1.0	7.3	88	7.9	110	46.35	0.3	8.8	26	7.1	17	16.09	0.5
G (13–14)	6.4	77	4.9	71		1.1	5.2	63	5.6	78		0.3	6.8	20	7.1	17		0.1
H (15–16)	6.0	72	4.2	60		1.4	4.1	50	3.6	51		0.4	5.4	16	4.1	10		0.0
I (17–18)	3.9	47	3.0	44		0.8	4.6	55	3.8	53		0.7	4.8	14	2.5	6		0.5
J (19–20)	3.0	36	2.7	39		0.3	3.0	36	2.1	30		0.9	3.1	9	2.9	7		1.0
K (21+)	10.3	124	6.3	91		2.4 †	11.0	132	5.4	76		3.4 †	8.5	25	5.8	14		0.9
Total number of ranges	1203		1445				1205		1401				294		241			
*p*	<0.001						<0.001						0.10					
ES	0.17						0.13						0.17					

* *p* < 0.05; † AR = |2.0|.

**Table 5 ijerph-16-04057-t005:** Differences in temporal parameters and range of strokes between men’s and women’s badminton players during long matches.

Variables	Set 1						Set 2						Set 3					
	Men		Women		*p*	ES	Men		Women		*p*	ES	Men		Women		*p*	ES
Temporal Parameters	M	SD	M	SD			M	SD	M	SD			M	SD	M	SD		
Set duration	25.96	3.11	23.37	3.57	0.34	0.77	25.25	4.70	24.62	4.06	0.75	0.14	27.13	3.63	30.19	6.81	0.23	0.56
Rally time	12.85	9.94	10.20	7.17	0.24	0.31	11.46	9.29	10.23	7.21	0.74	0.15	11.85	9.00	11.18	8.55	0.87	0.08
Real playing time	490.9	9.83	381.1	6.81	<0.01 *	2.11	433.3	8.85	381.2	6.85	<0.01 *	6.58	443.3	8.88	444.11	7.71	0.83	0.09
Rest time	26.98	12.47	25.46	12.90	0.53	0.12	27.71	14.24	27.35	12.17	0.95	0.03	30.92	16.12	31.54	18.98	0.94	0.04
Real rest time	976.5	12.14	900.2	12.08	<0.01 *	3.17	991.9	13.17	964.6	10.87	<0.01 *	2.26	1094.5	15.60	1189.8	15.79	<0.01 *	6.07
Density	0.48	0.32	0.41	0.22	0.57	0.32	0.42	0.27	0.38	0.20	0.71	0.17	0.40	0.26	0.36	0.21	0.71	0.17
Strokes per rally	13.47	10.06	9.68	7.08	0.34	0.43	11.95	9.48	9.84	7.20	0.58	0.25	12.45	9.12	10.86	8.29	0.69	0.18
Strokes per match	514.4	67.94	361.8	64.75	<0.01 *	2.23	451.7	96.98	366.7	70.23	0.04 *	1.00	465.6	73.20	431.5	131.5	0.48	0.32
Frequency of strokes	0.91	0.12	1.10	0.29	0.86	0.85	0.93	0.12	1.07	0.27	0.15	0.67	0.90	0.11	1.04	0.26	0.13	0.70
Number of rallies	38.20	2.25	37.36	3.20	0.51	0.30	37.80	3.08	37.27	3.07	0.70	0.17	37.40	2.67	39.73	4.84	0.20	0.59
Effective playing time (%)	33.45	2.18	29.73	2.18	<0.01 *	1.71	30.52	3.59	28.42	2.45	0.15	0.68	28.86	2.55	26.93	1.76	0.06	0.88
Effective rest time (%)	66.55	2.18	70.27	2.18	<0.01 *	1.71	69.48	3.59	71.58	2.45	0.15	0.68	71.14	2.55	73.07	1.76	0.06	0.88
Range of strokes	%	n	%	n	χ^2^	AR	%	n	%	n	χ^2^	AR	%	n	%	n	χ^2^	AR
A (1–2)	3.1	12	10.0	41		2.6 †	6.9	26	8.3	34		0.5	4.5	17	9.8	43		2.0
B (3–4)	14.4	55	16.5	68		0.5	12.7	48	17.6	72		1.2	12.8	48	13.7	60		0.2
C (5–6)	10.2	39	13.4	55		0.9	16.1	61	12.4	51		1.0	13.4	50	12.8	56		0.1
D (7–8)	12.6	48	13.4	55		0.2	10.3	39	16.1	66		1.5	12.0	45	13.3	58		0.3
E (9–10)	9.9	38	11.2	46		0.4	8.2	31	9.5	39		0.4	10.4	39	11.2	49		0.2
F (11–12)	7.1	27	8.0	33	44.44	0.3	11.6	44	10.0	41	22.75	0.5	7.2	27	8.7	38	15.10	0.5
G (13–14)	6.3	24	8.3	34		0.7	4.8	18	6.3	26		0.6	8.3	31	5.0	22		1.2
H (15–16)	7.3	28	5.4	22		0.8	5.8	22	4.4	18		0.8	5.9	22	4.8	21		0.5
I (17–18)	5.2	20	2.7	11		1.3	4.8	18	3.4	14		0.6	4.5	17	4.6	20		0.0
J (19–20)	3.7	14	3.4	14		0.1	2.9	11	3.4	14		0.3	3.5	13	3.0	13		0.3
K (21+)	20.2	77	7.8	32		3.3 †	15.9	60	8.5	35		2.1 †	17.4	65	13.0	57		1.0
Total number of ranges	382		411				378		410				374		437			
*p*	<0.001						0.01						0.13					
ES	0.24						0.17						0.14					

* *p* < 0.05; † AR = |2.0|.

**Table 6 ijerph-16-04057-t006:** Differences in point outcome between men’s and women’s badminton players according to match type.

Behaviour	Point Outcome	Men		Women		χ^2^	AR
	All Matches	n	%	n	%		
Serving	Winner	855	19.1	754	15.1		5.2 †
	Forced error	617	13.8	774	15.5		2.4 †
	Unforced error	747	16.7	854	17.1	44.26	0.5
Receiving	Winner	860	19.2	934	18.7		0.6
	Forced error	586	13.1	597	12.0		1.7
	Unforced error	810	18.1	1077	21.6		4.2 †
	Total number of points	4475		4990			
	P	<0.001 *					
	ES	0.07					
	Short matches	%	n	%	n	χ^2^	AR
Serving	Winner	127	19.9	120	18.6		0.6
Serving Receiving	Forced error	83	13.0	83	12.9		0.1
	Unforced error	92	14.4	127	19.7	7.62	2.5 †
	Winner	124	19.4	106	16.4		1.4
Receiving	Forced error	87	13.6	79	12.2		0.7
	Unforced error	126	19.7	130	20.2		0.2
	Total number of points	639	645				
	*p*	0.178					
	ES	0.07					
	Regular matches	%	n	%	n	χ^2^	AR
Serving	Winner	514	19.0	462	15.0		4.1 †
Serving Receiving	Forced error	383	14.2	482	15.6		1.5
	Unforced error	474	17.5	535	17.3	35.64	0.2
	Winner	504	18.7	563	18.2		0.4
Receiving	Forced error	357	13.2	360	11.7		1.8
	Unforced error	470	17.4	685	22.2		4.6 †
	Total number of points	2702	3087				
	*p*	<0.001 *					
	ES	0.08					
	Long matches	%	n	%	n	χ^2^	AR
Serving	Winner	214	18.9	172	13.7		3.5 †
Serving Receiving	Forced error	151	13.3	209	16.6		2.3 †
	Unforced error	181	16.0	192	15.3	15.74	0.5
	Winner	232	20.5	265	21.1		0.4
Receiving	Forced error	142	12.5	158	12.6		0.0
	Unforced error	214	18.9	262	20.8		1.2
	Total number of points	1134	1258				
	*p*	0.008 *					
	ES	0.08					

* *p* < 0.05; † AR = |2.0|.

**Table 7 ijerph-16-04057-t007:** Differences in temporal parameters and range of strokes between men’s and women’s badminton players during long matches.

Behaviour	Point Outcome	Set 1						Set 2						Set 3					
		Men		Women		χ^2^	AR	Men		Women		χ^2^	AR	Men		Women		χ^2^	AR
	Short Matches	n	%	n	%			n	%	n	%			n	%	n	%		
Serving	Winner	68	21.8	59	19.2		0.8	59	18.0	61	18.1		0.0						
	Forced error	41	13.1	38	12.3		0.3	42	12.8	45	13.4		0.2						
	Unforced error	46	14.7	59	19.2	3.37	1.5	46	14.1	68	20.2	10.61	2.1						
Receiving	Winner	48	15.4	53	17.2		0.6	76	23.2	53	15.7		2.4						
	Forced error	41	13.1	41	13.3		0.1	46	14.1	38	11.3		1.1						
	Unforced error	68	21.8	58	18.8		0.9	58	17.7	72	21.4		1.2						
	Total number of points	312		308				327		337									
	*p*	0.638						0.061											
	ES	0.07						0.13											
	**Regular matches**	n	%	n	%	χ^2^	AR	n	%	n	%	χ^2^	AR	n	%	n	%	χ^2^	AR
Serving	Winner	236	19.6	195	13.5		4.2 †	224	18.6	227	16.2		1.6	54	18.4	40	16.6		0.5
	Forced error	152	12.6	249	17.2		3.3 †	187	15.5	205	14.6		0.6	44	15.0	28	11.6		1.1
	Unforced error	222	18.5	247	17.1	31.55	0.9	196	16.3	253	18.1	11.27	1.2	56	19.0	35	14.5	11.22	1.4
Receiving	Winner	227	18.9	266	18.4		0.3	224	18.6	251	17.9		0.4	53	18.0	46	19.1		0.3
	Forced error	149	12.4	165	11.4		0.8	166	13.8	166	11.8		1.5	42	14.3	29	12.0		0.8
	Unforced error	217	18.0	323	22.4		2.7†	208	17.3	299	21.3		2.6†	45	15.3	63	26.1		3.1 †
	Total number of points	1203		1445				1205		1401				294		241			
	*p*	<0.001 *						0.046 *						0.049 *					
	ES	0.11						0.07						0.15					
	**Long matches**	n	%	n	%	χ^2^	AR	n	%	n	%	χ^2^	AR	n	%	n	%	χ^2^	AR
Serving	Winner	72	18.8	61	14.8		1.5	70	18.5	55	13.4		2.0	72	19.3	56	12.8		2.5
	Forced error	48	12.6	69	16.8		1.7	55	14.6	65	15.9		0.5	48	12.8	75	17.2		1.7
	Unforced error	63	16.5	62	15.1	10.77	0.5	64	16.9	67	16.3	10.81	0.2	54	14.4	63	14.4	9.70	0.0
Receiving	Winner	62	16.2	94	22.9		2.4 †	91	24.1	80	19.5		1.6	79	21.1	91	20.8		0.1
	Forced error	57	14.9	51	12.4		1.0	33	8.7	53	12.9		1.9	52	13.9	54	12.4		0.7
	Unforced error	80	20.9	74	18.0		1.0	65	17.2	90	22.0		1.7	69	18.4	98	22.4		1.4
	Total number of points	382		411				378		410				374		437			
	*p*	0.049 *						0.06						0.08					
	ES	0.12						0.12						0.11					

* *p* < 0.05; † AR = |2.0|.

**Table 8 ijerph-16-04057-t008:** Main differences between sexes of temporal and notational variables according to match type and set.

Variables	All Matches	Short Matches	Regular Matches	Long Matches
All	Set 1	Set 2	All	Set 1	Set 2	Set 3	All	Set 1	Set 2	Set 3
Temporal parameters												
Set duration								↑M				
Rally time	↑M				↑M				↑M			
Real playing time			↑M	↑M	↑M	↑M	↑M	↑M	↑M	↑M	↑M	
Rest time	↑M	↑M										
Real rest time			↑M	↑M			↑M	↑M		↑M	↑M	↑W
Density	↑M	↑W			↑M				↑W			
Strokes per rally	↑M				↑M				↑M			
Strokes per match/set	↑M		↑M		↑M	↑M	↑M	↑M	↑M		↑M	
Frequency between strokes	↑W	↑W	↑W		↑M				↑W			
Number of rallies			↑M						↑W			
Effective playing time (%)			↑M						↑M	↑M		
Effective rest time (%)		↑ M	↑W						↑W	↑W		
Range of strokes												
A (1–2)	↑W				↑W	↑W			↑W	↑W		
B (2–3)	↑W											
H (15–16)	↑M											
I (17–18)	↑M											
K (>20)	↑M	↑M			↑M	↑M	↑M		↑M	↑M		
Point outcome												
Serving winners	↑M				↑M	↑M			↑M			
Serving forced errors	↑W								↑W			
Serving unforced errors		↑W				↑W						
Receiving winners										↑M		
Receiving unforced errors	↑W				↑W	↑W	↑W	↑W

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
