# Peer review of "Performance Differences of Temporal Parameters and Point Outcome Between Elite Men’s and Women’s Badminton Players According to Match-Related Contexts"

_ijerph, 2019, doi:10.3390/ijerph16214057_

Round 1
Reviewer 1 Report
The Introduction is well structured. The authors present a theoretical framework supported by relevant and current literature to explain the importance of the study. The objective and hypotheses are well specified and located in the document and correspond to what is described in the Abstract, but for better compression abstract have to be rewritten. Please include in the title the sport speciality of the subjects
Method section: The way of recruiting the participants, as well as the methodology of the study, seem adequate to respond to the hypotheses raised. The universe under study is well represented.
Results section: They are presented in an orderly manner and are interesting for the field of study.
Discussion section: In this section, the authors explain the data obtained and their relationship with the variables analyzed. Is necessary to explain the differences observed. include a practical application section and in the conclusion section just respond to the aims of the study
Author Response
Reviewer 1
REV.- The Introduction is well structured. The authors present a theoretical framework supported by relevant and current literature to explain the importance of the study. The objective and hypotheses are well specified and located in the document and correspond to what is described in the Abstract, but for better compression abstract have to be rewritten.
AUT.- Thanks for your revision and constructive comments to improve the manuscript. The abstract section was rewritten to clarify the meaning of the section (see L17-L33).
REV.- Please include in the title the sport specialty of the subjects
AUT.- Thanks for this suggestion, the sport of “badminton” was included in the title (L4).
REV.- Method section: The way of recruiting the participants, as well as the methodology of the study, seem adequate to respond to the hypotheses raised. The universe under study is well represented.
AUT.- Thanks for your positive feedback.
REV.- Results section: They are presented in an orderly manner and are interesting for the field of study.
AUT.- Thanks for your positive feedback.
REV.- Discussion section: In this section, the authors explain the data obtained and their relationship with the variables analyzed. Is necessary to explain the differences observed. Include a practical application section and in the conclusion section just respond to the aims of the study
AUT.- Thanks for your suggestions. The main differences between genders were included in the Discussion section (P7-8, second and third paragraphs of discussion section). The practical applications of the study were included and the conclusion section was udpated (P8-9, L282-302).
Reviewer 2 Report
The work is good, it's well written, methodologically correct.
That its weak point is the literature review, which, being good, is improvable.
There are few works on the specific theme of badminton, but there are some studies in racket sports that could complement the paper.
Possibly a second issue that could be improved in the presentation
of the work is what suggestions the authors have regarding the
implications in the training of the results obtained. In what way
they should adapt.
Author Response
Reviewer 2
REV.- The work is good, it's well written, methodologically correct.
AUT.- Thanks for your revision and constructive comments to improve the manuscript.
REV.- That its weak point is the literature review, which, being good, is improvable.
AUT.- The article was revised accordingly with addition of references focused on badminton and racket sports (references 5,6,11, 27,28 focused on badminton, references 7 and 8 in performance analysis in racket sports, 15-17 in tennis, 18,20 table tennis, 21-22 squash and 23-24 paddle).
REV.- There are few works on the specific theme of badminton, but there are some studies in racket sports that could complement the paper.
AUT.- The article was revised accordingly with addition of references focused on badminton and racket sports (references 5,6,11, 27,28 focused on badminton, references 7 and 8 in performance analysis in racket sports, 15-17 in tennis, 18,20 table tennis, 21-22 squash and 23-24 paddle).
REV.- Possibly a second issue that could be improved in the presentation
of the work is what suggestions the authors have regarding the
implications in the training of the results obtained. In what way
they should adapt.
AUT.- Thanks for your suggestion. The practical applications of the study were included (P8-9, L282-302).
Round 2
Reviewer 1 Report
congratulation to the author for the nice work conducted